# Potential health and economic impacts of dexamethasone treatment for patients with COVID-19

Ricardo Águas[1], Adam Mahdi[2], Rima Shretta [3], Peter Horby[4], Martin Landray [5], Lisa White [3✉] & the CoMo Consortium*

Dexamethasone can reduce mortality in hospitalised COVID-19 patients needing oxygen and ventilation by 18% and 36%, respectively. Here, we estimate the potential number of lives saved and life years gained if this treatment were to be rolled out in the UK and globally, as well as the cost-effectiveness of implementing this intervention. Assuming SARS-CoV-2 exposure levels of 5% to 15%, we estimate that, for the UK, approximately 12,000 (4,250 - 27,000) lives could be saved between July and December 2020. Assuming that dexamethasone has a similar effect size in settings where access to oxygen therapies is limited, this would translate into approximately 650,000 (240,000 - 1,400,000) lives saved globally over the same time period. If dexamethasone acts differently in these settings, the impact could be less than half of this value. To estimate the full potential of dexamethasone in the global fight against COVID-19, it is essential to perform clinical research in settings with limited access to oxygen and/or ventilators, for example in low- and middle-income countries.

[1] Mahidol-Oxford Tropical Medicine Research Unit, Nuffield Department of Medicine, University of Oxford, Oxford, UK. [2] Institute of Biomedical Engineering, Department of Engineering Science, University of Oxford, Oxford, UK. [3] Big Data Institute, Li Ka Shing Centre for Health Information and Discovery, Nuffield Department of Medicine, University of Oxford, Oxford, UK. [4] Centre for Tropical Medicine and Global Health, Nuffield Department of Medicine, University of Oxford, Oxford, UK. [5] Medical Research Council Population Health Research Unit at the University of Oxford, Nuffield Department of Population Health, Oxford, UK. *A list of authors and their affiliations appears at the end of the paper. ✉email: lisa.white@ndm.ox.ac.uk

Coronavirus disease 2019 (COVID-19) emerged in late 2019 and is either asymptomatic or causes only mild symptoms in most individuals[1]. However, a significant number of individuals, especially among the elderly, develop a more severe form of the disease and require hospital care. A further subset of these patients requires oxygen therapy or ventilatory assistance, creating a high demand for long-term hospital care in intensive care units (ICUs). In the UK, between 6 February and 18 April 2020, the case fatality rate among those admitted to hospital with COVID-19 was more than 26%; this increased to more than 37% in patients who required mechanical ventilation[2]. An extensive search for potential drugs with which to treat COVID-19 is underway. Dexamethasone has emerged as a standout therapeutic candidate[3], reducing mortality in hospitalised COVID-19 patients needing oxygen and ventilation by 18% and 36%, respectively[4]. Thus, the adoption of a dexamethasone treatment protocol for patients requiring respiratory support could potentially lead to a significant number of lives saved over the course of the pandemic between July and December 2020.

Lockdown measures have been eased around the world and at various times, creating concern about potential resurgences in COVID-19 activity[5]. As stringent transmission-reducing interventions are lifted, the expectation is that further epidemic peaks become inevitable, and with them increases in mortality. Dexamethasone is an affordable medicine, available as a generic product, that has been routinely used in hospital settings globally. It is, therefore, assuming global availability is maintained[6], perfectly placed as a candidate standard therapeutic option for COVID-19 patients with respiratory distress, which could reduce the future mortality burden of this disease in a cost-effective way.

Estimates of the future burden of COVID-19, and therefore the impact of any treatments for COVID-19, are highly dependent both on viral and human behavioural factors. Both are riddled with uncertainty and are changing as the pandemic evolves. Several biological parameters describing how the virus transmits between individuals and how infectivity progresses following infection remain elusive. A critical metric that reflects both how transmissible and lethal the virus can be is the infection fatality ratio (IFR). Until very recently, obtaining a clear picture of how many people have been infected was only feasible in small, well-contained outbreaks, such as those on cruise ships[7]. Even in those instances, the narrow time window for good polymerase chain reaction (PCR) testing sensitivity meant even the best estimates for IFR were changing on a weekly basis. Establishing the relationship between deaths and infections is crucial to enable reliable mortality burden predictions for the next wave of the epidemic.

Since May 2020, serological studies have been conducted, in several countries, to measure the proportion of the population that has been exposed to the virus, including in the UK[8]. Up to 28 June (week 27) 2020, we know that at least 7% of the English population had been exposed to COVID-19, while around 40,000 patients had died from the disease[9]. Taking the reported reduction in mortality risk due to dexamethasone treatment[4] and assuming that 59% of hospital patients receive oxygen and 17% are ventilated[4], we conclude that dexamethasone can reduce mortality by 16.75% in a hospital setting, according to this simple calculation: $[(0.18 \times 0.59) + (0.36 \times 0.17)] = 0.1675$.

Given a reasonable IFR estimate, we can project different SARS-COV-2 transmission progression scenarios between June 2020 and January 2021, along with their respective mortality burdens, notwithstanding a great deal of uncertainty around what future social distancing measures will be taken and how the population will adhere to those measures.

Here, we describe a simple and transparent approach to determine the potential benefits of implementing a standard COVID-19 treatment protocol with dexamethasone in terms of lives saved, life-years gained, and cost per life saved. In simple terms, given a country's population age structure, we use age-dependent estimates for the relationships between infection and mortality and infection and hospitalisation to extrapolate the expected number of hospital admissions and COVID-19 deaths given a specific projected number of infections between June 2020 and January 2021. Assumptions about oxygen treatment requirements and expected probabilities of mortality given the use of dexamethasone are clearly stated and can be altered to explore alternative options for these values.

## Results

We consider the treatment of patients hospitalised with COVID-19 disease, as defined in[3], as when a patient presents with (i) typical symptoms (e.g. fatigue with fever and muscle pain, or respiratory illness with cough and shortness of breath); and (ii) compatible chest X-ray findings (consolidation or ground-glass shadowing); and (iii) alternative causes have been considered unlikely or excluded (e.g. heart failure, influenza). However, the diagnosis remains mainly based on clinical symptoms and is ultimately at the managing doctor's discretion.

Working from the population age distribution in the UK as reported by the UN 2019 Revision of World Population Prospects[10], we assume that among those who are infected, the proportion hospitalised is as estimated for the French population[11]. We also assume that the hospitalised case fatality ratio by age is correlated with that estimated for the French population[11]. More specifically, we developed a hospitalised patient treatment pathway, similar to the one in[12], which can be distilled into a decision tree algorithm (Fig. 1) driven by the parameters listed in Table 1 and Supplementary Table 1.

This algorithm can straightforwardly translate a projected number of total infections into an expected number of hospitalisations by age group. For a range of likely exposure levels between July and December 2020, we can estimate the respective expected mortality, as was done in[13], by applying the probabilities in Table 1 and following the branching processes in Fig. 1. We make this estimate under two scenarios: (1) patients do not receive dexamethasone; (2) patients receive dexamethasone if they meet the criteria for the treatment as described in[4]. The number of potential lives saved by adopting dexamethasone as a standard COVID-19 treatment is simply the difference between these figures. We estimate that, for the UK, ~12,000 (4250–27,000 90% confidence interval) lives could be saved over the period from July to December 2020, under perfect access to treatment.

However, whilst in the UK it is reasonable to assume patients will have near-perfect access to whatever treatment they require, including dexamethasone, that assumption is unrealistic for many low- and middle-income countries (LMICs). Of note, in scenarios where access to oxygen therapy or ventilatory support is limited, the effect of dexamethasone on patient outcome is unknown. It is therefore essential to perform clinical research in LMICs, where access to oxygen and ventilators is limited, to estimate the full potential of dexamethasone in the global fight against COVID-19.

To address this uncertainty, we simulated the use of dexamethasone if a patient meets the oxygen prescribing target described in[14], whether or not oxygen is available. Whilst the impact of dexamethasone on patients who require but cannot access supplemental oxygen or invasive mechanical ventilation is not known, it is reasonable to assume that some mortality benefit would arise from the reduction of inflammation and consequent decrease in the probability of progressing to respiratory failure. We estimated the potential number of lives saved under two assumptions: (1) dexamethasone has the same relative impact on mortality in patients who do not receive the respiratory support

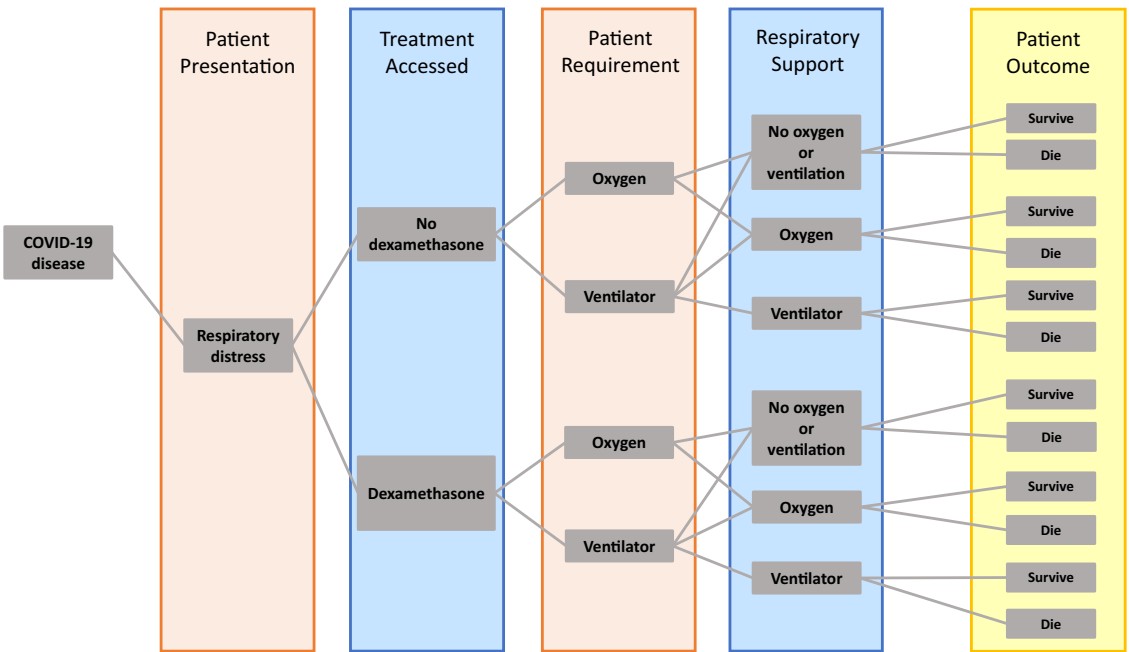

**Fig. 1 Patient pathway diagram for the calculation of COVID-19 mortality for a given exposure level.** We assign two categories of patients who are eligible for dexamethasone treatment. Those who require non-invasive oxygen (but never require ventilation) and those who ultimately require mechanical ventilation (where we assume their use of non-invasive oxygen during this pathway is negligible in terms of time and cost). Each group has two survival rates associated with them, depending on whether they receive dexamethasone treatment or not. A percentage of patients will not progress to requiring ventilation if they receive dexamethasone treatment early, and this is reflected in the "Oxygen":"Ventilator" ratio of the "Dexamethasone" branch of the pathway.

**Table 1 Definitions of the values used for the calculation and, where relevant, the ranges of values used for the sensitivity analysis.**

| Definition | Symbol | Value | Lower | Upper | Source |
|---|---|---|---|---|---|
| The proportion of the population expected to be exposed from July to December 2020 | $\alpha$ | 0.07 | 0.05 | 0.15 | [9] |
| Level of access to appropriate respiratory support (as a proportion) | $\kappa$ | 1 | 0.5 | 1 | – |
| Increase in fatality rate if no access to appropriate respiratory support | $\psi$ | 1 | 1 | 2 | – |
| Probability of hospitalised patient needing oxygen (either non-invasive or ventilator) | $p_{O_2}$ | 0.76 | 0.2 | 0.9 | [4] |
| Probability of a hospitalised patient who receives oxygen needing a ventilator if on dexamethasone | $p_{Vd}$ | 0.22*0.87 | 0.04 | 0.5 | [4] |
| Probability of an infected patient who receives oxygen needing a ventilator if not on dexamethasone | $p_V$ | 0.22 | 0.05 | 0.6 | [4] |
| Maximum death probability for a hospitalised infection | $\delta_H$ | 0.3 | – | – | [2] |
| Maximum death probability for a hospitalised infection requiring non-invasive oxygen | $\delta_{O_2}$ | 0.4 | – | – | [2] |
| Maximum death probability for a hospitalised infection requiring a ventilator | $\delta_V$ | 0.66 | – | – | [2] |
| Reduction in mortality for patients requiring non-invasive oxygen if on dexamethasone | $\epsilon_{O_2}$ | 0.2 | – | – | [4] |
| Reduction in mortality for patients requiring a ventilator if on dexamethasone | $\epsilon_V$ | 0.36 | – | – | [4] |
| Incremental number of days spent in a general ward bed for a survivor that required non-invasive oxygen | $ids_{o_2}$ | 4.5 | 2 | 7 | [20] |
| Incremental number of days spent in a general ward bed for a survivor that required ventilation | $ids_v$ | 5 | 3 | 8 | [20] |
| Incremental number of days spent in an ICU bed for a survivor that required ventilation | $idc_v$ | −1 | −2 | 3 | [20] |

they require as in those who do; (2) dexamethasone has no impact on the patient outcome if the patient does not receive the oxygen or ventilation they need. We appreciate these are extreme assumptions but in the absence of information we prefer to offer the widest possible interval of plausibility. When including access to respiratory support as a covariate in the sensitivity analysis, we predict an estimated ~650,000 (240,000–1,400,000 90% confidence interval) potential lives saved worldwide under the first assumption and 390,000 (130,000–1,000,000 90% confidence interval) lives saved worldwide under the second assumption.

Dexamethasone is an affordable drug that has been on the market for many years[15]. Its use can substantially alter outcomes for patients with COVID-19 and by doing so will impact hospital occupancy management and the overall cost of treatment. Here,

we also explore the economic ramifications of adopting dexamethasone as a default option for treating patients with COVID-19 and provide metrics for the cost-effectiveness of this conceivable health policy change in the UK. To accomplish this, we procured daily hospital patient costs per treatment given as well as data on length of hospital stay for patients with different hospital pathways and disease outcomes—see "Methods" for more details. According to National Institute for Health and Care Excellence (NICE) directives, interventions with an incremental cost-effectiveness ratio (ICER) of <£20,000 per quality-adjusted life-year (QALY) or life-year gained are considered to be cost effective[16].

In the UK, we estimate a total incremental cost of £85,000,000 (£6,000,000–£330,000,000 90% confidence interval) from July to

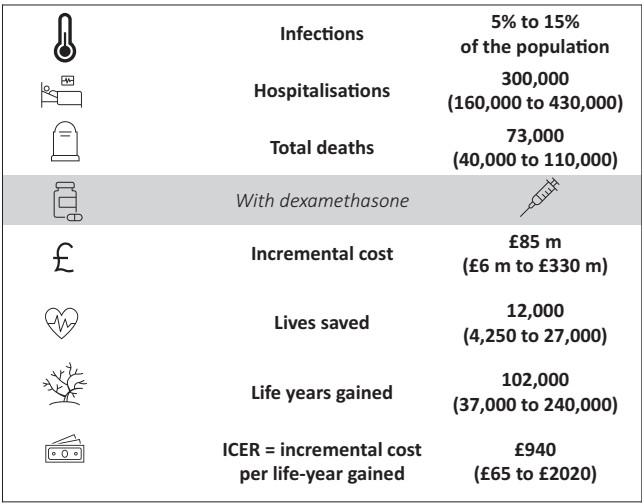

| | | |
|---|---|---|
| | Infections | 5% to 15% of the population |
| | Hospitalisations | 300,000 (160,000 to 430,000) |
| | Total deaths | 73,000 (40,000 to 110,000) |
| | *With dexamethasone* | |
| | Incremental cost | £85 m (£6 m to £330 m) |
| | Lives saved | 12,000 (4,250 to 27,000) |
| | Life years gained | 102,000 (37,000 to 240,000) |
| | ICER = incremental cost per life-year gained | £940 (£65 to £2020) |

**Fig. 2 Expected impact of dexamethasone in the UK from July to December 2020 for the range of scenarios explored.** The value and range quoted for each outcome represent the median and 5th and 95th percentiles of the sensitivity analysis outcomes, i.e., 90% double-sided confidence intervals. The letter m represents million.

December 2020, which equates to £8,200 (£650–17,500 90% confidence interval) per life saved and £940 (£65–£2020 90% confidence interval) per life-year gained, making dexamethasone treatment a clearly cost-effective option (Fig. 2). A cost-effectiveness acceptability curve (CEAC) was constructed to quantify and graphically represent uncertainty in the economic evaluation. The CEAC (Fig. 3A) shows that the decision uncertainty surrounding the adoption of dexamethasone, at a maximum acceptable ceiling ratio of £2,020 per life-year gained, is 5% (given the probability that dexamethasone is cost-effective is 0.95) for the default analysis where there is perfect access to treatment.

We have refrained from presenting these estimates for the rest of the world due to the large variance in hospital patient management, hospitalisation costs, and access to treatment in other countries, and especially in LMICs. We did, however, perform an exploratory analysis to ascertain the sensitivity of our ICER estimates to assumptions regarding the probability of receiving oxygen therapy when needed and the efficacy of dexamethasone when not receiving oxygen (Fig. 3b). The results show non-significant differences in the median ICERs for the explored parameter ranges.

## Discussion

Dexamethasone is a globally accessible, existing treatment that can be highly cost effective if given to hospitalised COVID-19 patients requiring oxygen therapy. In the UK setting specifically, dexamethasone is estimated to save up to 27,000 lives from July to December 2020 at a cost of up to £2020 per life-year gained, which outshines the recently implemented HPV vaccination ICER[17]. In high-income countries with an ageing population, where access to respiratory support is not expected to be an issue, dexamethasone should promptly be adopted as the standard treatment for patients with respiratory distress. This treatment, if given in accordance with the current guidance on patient eligibility for oxygen therapy, even where access to oxygen is limited, could save hundreds of thousands of lives from July to December 2020 of the pandemic. The findings presented here provide a foundation for discussions around the prospective adoption of dexamethasone treatment by LMICs, where access to oxygen therapy might be limited. In countries where patients requiring oxygen therapy are expected to have a lower than 25% chance of

receiving it, clinical studies should be undertaken to determine whether dexamethasone is at least 50% efficacious, thus assuring the cost effectiveness of dexamethasone treatment. Those studies should be informed by a comprehensive expected value of partially perfect information (EVPPI) analysis, building on the results presented here, to inform the most appropriate sample sizes.

In settings where social distancing and other non-pharmaceutical interventions are still being deployed and in other settings where these measures are becoming untenable, dexamethasone could serve both to reduce mortality and mitigate the burden on health systems.

## Methods

We calculate the population expected to be exposed in each age class by multiplying the proportion of population expected to be exposed from July to December 2020, $\alpha$, (Table 1) by the number of people in each age group, $n$ (Supplementary Table 1). We obtain the number of people expected to require hospitalisation by multiplying the previous figure by the age-structured proportion of all exposures leading to hospitalisation, $p_H$, (Supplementary Table 1). We then multiply this vector by various combinations of probabilities depending on the within-hospital patient pathway (Fig. 1), structured by normalised hospitalisation fatality rate, $p_F$, (Supplementary Table 1) as follows:

- Expected deaths in hospitalised patients that do not require oxygen (either non-invasive or ventilator) in each age category:

$$y_H = \alpha n p_H p_F \left(1 - p_{O_2}\right)[\kappa + (1 - \kappa)(1 + \psi)]\delta_H \quad (1)$$

- Expected deaths in hospitalised patients who require non-invasive oxygen and receive dexamethasone in each age category:

$$y_{O_2} = (1 - \epsilon_{O_2})\alpha n p_H p_F p_{O_2}[\kappa(1 - p_V) + (1 - \kappa)(1 - p_V)(1 + \psi)]\delta_{O_2} \quad (2)$$

- Expected deaths in hospitalised patients who require mechanical ventilation and receive dexamethasone in each age category:

$$y_V = (1 - \epsilon_V)\alpha n p_H p_F p_{O_2} p_{Vd}\left[\kappa^2 + (1 - \kappa)(1 + \psi) + \kappa(1 - \kappa)(1 + \psi)\right]\delta_V \quad (3)$$

Then, the expected number of deaths in patients in each age category with severe respiratory distress that receive dexamethasone is given by:

$$y_D = y_H + y_{O_2} + y_V \quad (4)$$

The expected number of deaths in each age category in the same patients if they do not receive dexamethasone is:

$$y_0 = y_H + \frac{y_v p_V}{(1 - \epsilon_V)p_{Vd}} + \frac{y_{O_2}}{(1 - \epsilon_{O_2})} \quad (5)$$

The potential lives saved, summed over all age categories, is then given by $\sum y_L$, where:

$$\sum y_L = \sum (y_0 - y_D) \quad (6)$$

The model follows the disease pathway of COVID-19-infected patients with severe respiratory illness requiring hospitalisation, non-invasive oxygen and mechanical ventilation. Depending on the strategies under investigation, patients may or may not be further split by type of management (general ward versus ICU) and use of medication (e.g. dexamethasone). In short, a proportion $p_{O_2}$ of individuals admitted to hospital with respiratory distress will require some form of oxygen. A subset of those, $p_{Vd}$, will need mechanical ventilation. If people have perfect access to the treatment they require, their outcome is determined by a treatment specific, age-dependent probability of death. This is calculated by multiplying the age-dependent mortality modulation vector $p_F$ (Supplementary Table 1) by the appropriate treatment requirement death probabilities in Table 1. If access to oxygen and/or ventilation is not perfect, the patient will not receive the most appropriate treatment, which will be reflected in an increase in mortality probability, given by $\psi$. Prompt treatment with dexamethasone is assumed to reduce the need for patient ventilation by 13%, $p_{Vd}$.

The costing analysis makes use of the simple state transition modelling framework described above (and illustrated in Fig. 1). The analysis takes a provider (health system) perspective. Costs are expressed in 2020 prices, and no discounting on costs is undertaken given that all costs within the analyses are incurred within a short period of time. For the years of life lost (YLL), a standard discrete-time discounting approach was used to estimate the present value of a life saved. A discrete-time discount factor, $1/(1 + r)t$ was applied, where r is the discount rate and t the number of years[18].

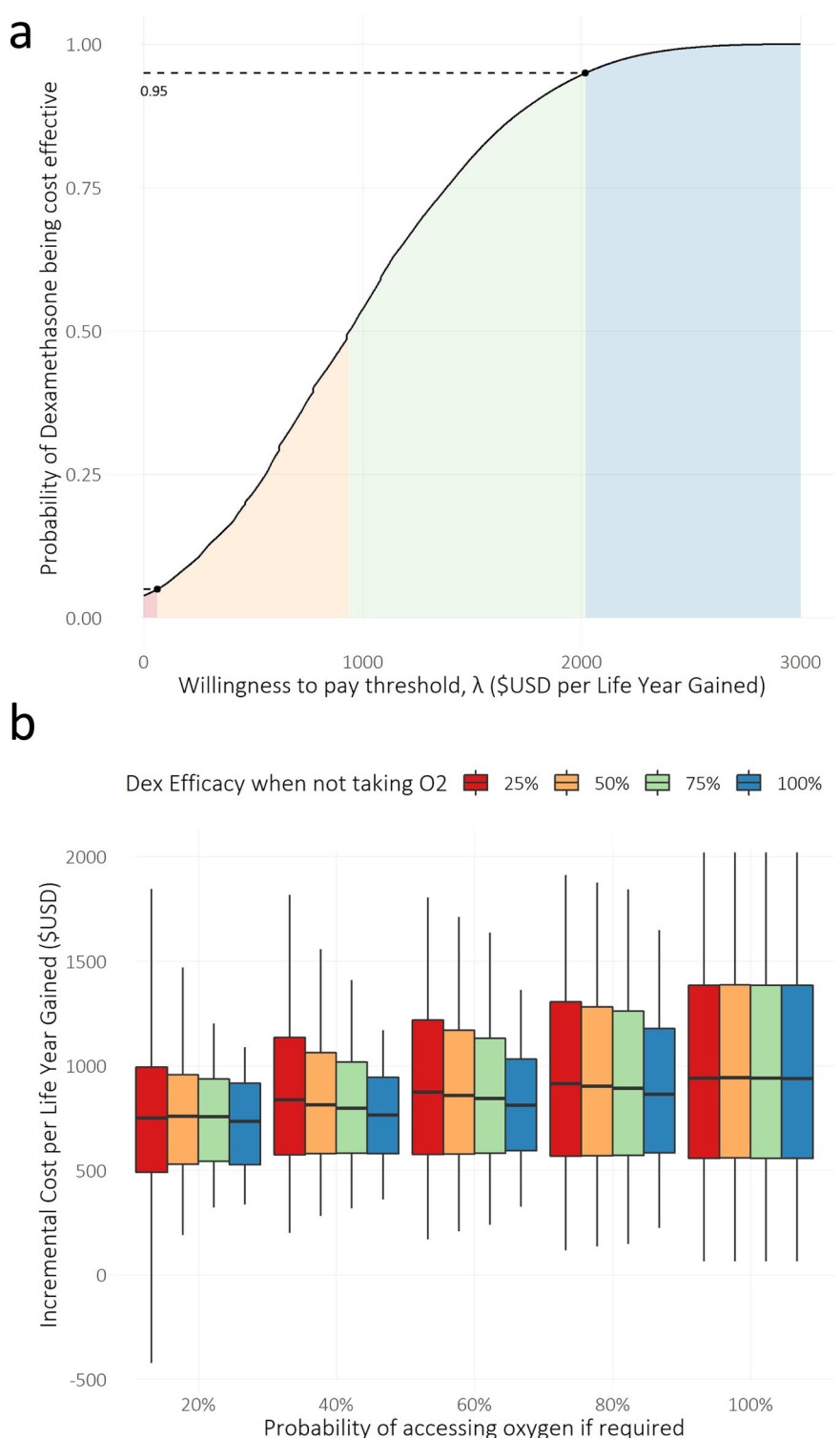

**Fig. 3 Uncertainty regarding access to oxygen therapy. a** A cost-effectiveness acceptability curve constructed to quantify and graphically represent uncertainty in the economic evaluation. The *x*-axis represents ĸ, the cost-effectiveness threshold, while the *y*-axis reflects the estimated probability of dexamethasone being cost-effective. We highlighted the probability of dexamethasone being cost effective at the ICER threshold value corresponding to the upper end of the 90% confidence interval for the ICER obtained in the sensitivity analysis. The shaded areas are defined by the ICER threshold values corresponding to the mean and 90% confidence intervals obtained in the probabilistic sensitivity analysis: red, < 5% of the distribution area; orange, 5–50%; green, 50–95%; blue, 95–100%. **b** Sensitivity of the evaluation of dexamethasone's cost effectiveness to assumptions in access to oxygen therapy and efficacy of dexamethasone on people that do not receive oxygen. Boxplots were drawn to represent the distribution of 1 million predicted incremental cost per life-years gained values for each combination of dexamethasone efficacy when not receiving oxygen and the probability of receiving oxygen treatment if required. The middle line is the median, the lower and upper hinges correspond to the first and third quartiles, and the whiskers extend to the 5th and 95th percentiles.

The models include two absorbing states: recovered and dead. The former is assumed to have no morbidity loss (i.e. patients return to their pre-COVID-19 health state upon recovery) while the latter captures the YLL for a person dying of COVID-19. The YLL associated with death is taken from secondary data and is based on estimated life expectancy for UK citizens in each of the five-year age groups[19].

From the patient pathway model in Fig. 1, we can extrapolate hospital utilisation, including the proportion of hospitalised individuals occupying ICU versus surge beds. To calculate hospitalisation costs per patient hospital pathway, we then need to know the length of stay per patient in each type of hospital bed. We assume that patients requiring oxygen alone will only occupy surge beds during their stay, regardless of the outcome. However, patients that do recover spend on average 4.5 more days in hospital than those who die[20,21]. Patients needing mechanical ventilation will occupy ICU beds for an average of 7 days if they survive and then spend an extra 5 days on average in a surge bed[20]. Ventilated patients who eventually die typically spend 8 days in an ICU bed. We then use the incremental number of days spent in the hospital, together with unit cost data per inpatient day in surge beds and intensive care beds, taken from the NHS National Tariff[22], to inform the total incremental cost of lives saved and life-years gained. The costs used are provided in Supplementary Table 2. To assess the cost-effectiveness of dexamethasone treatment, we need to compute the additional cost of hospital treatment including dexamethasone as compared with the cost of treatment without dexamethasone, known as the ICER. To calculate the ICER, the cost of providing dexamethasone is subtracted from the cost of treatment without dexamethasone and divided by the difference in YLL for the two treatment arms (no dexamethasone vs dexamethasone). The difference between the YLL for the two treatment arms then becomes the years of life gained (YLG).

$$\text{ICER} = \frac{\text{incremental cost of providing dexamethasone}}{\text{life years gained}} \quad (7)$$

To reflect the uncertainty inherent to the estimated numbers of deaths prevented, life-years gained, total incremental cost, the incremental cost per life saved, and incremental cost per life-year gained, we perform sensitivity analyses through Latin-hypercube sampling. Simply put, we iteratively calculate our interest outcome variables, where at each iteration we sample a value for selected input values from uniform distributions with set intervals, with ranges given in Table 1. For all sensitivity analyses we perform 1 million iterations and present the results as median and 5th and 95th percentiles, i.e. the 90% double-sided confidence intervals. Throughout the manuscript, we present epidemiological and costing estimates calculated when assuming the infection hospitalisation ratio (IHR) and the hospitalisation fatality ratio (HFR) in the UK resembles that estimated from French data[11]. Ideally, we would use UK age-dependent serology data to inform a UK-specific IHR. Unfortunately, those data are not currently available so we must rely on estimates/data from other countries. While we believe the French estimates are a good reflection of the UK reality, we re-calculate all estimates (Supplementary Table 3), assuming IHR and HFR to follow age patterns similar to those measured in Spain[23,24] (Supplementary Table 4).

CEACs summarize the impact of uncertainty on the result of economic evaluations. CEACs help decision-makers to understand the uncertainty associated with making the decision to adopt dexamethasone as a life-saving strategy for severe COVID19. Given the distributions in outcome variables obtained from the probabilistic sensitivity analysis described above, we used cost and life-years gained sampled pairs to construct a CEAC. Thus, uncertainty is characterized by estimating the probability that an option is cost-effective at different levels of the cost-effectiveness threshold. The probability of dexamethasone treatment being cost-effective is then equivalent to the proportion of the 1 million iterations for which dexamethasone had the highest net benefit, compared with ICER threshold values ranging from 0 to 3000 $USD per life-year gained (Fig. 3).

Following the sensitivity analysis described above, where the input parameters in Table 1 were explored to extrapolate ranges for predicted numbers of deaths prevented, life-years gained, total incremental cost, the incremental cost per life saved, and incremental cost per life-year gained, we extended the list of input parameters to explore adding the probability of accessing oxygen therapy if required and the relative efficacy of dexamethasone in patients not receiving oxygen, compared with guideline treatment efficacy. The resulting ICER ranges are illustrated in Fig. 3B.

**Reporting summary**. Further information on research design is available in the Nature Research Reporting Summary linked to this article.

## Data availability
Data sharing is not applicable to this article as no datasets were generated or analysed during the current study. The sources for the estimates used in the analyses are listed in Table 1.

## Code availability
The code used to perform the sensitivity analysis can be found at https://github.com/ATOME-MORU/dexamethasone-/[25] and is published under doi: 10.5281/zenodo.4302475. Anyone is free to download, edit and redistribute this code, within the

terms of the Attribution-NonCommercial 4.0 International license. Figure 3 can be reproduced by running the code provided in the github repository.

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

## Acknowledgements

We thank all members of the COVID-19 International Modelling Consortium and their collaborative partners. This work was supported by the COVID-19 Research Response Fund, managed by the Medical Sciences Division, University of Oxford. L.J.W. is supported by the Li Ka Shing Foundation. R.A. acknowledges funding from the Bill and Melinda Gates Foundation (OPP1193472).

## Author contributions

R.Á., R.S. and L.W. contributed equally to the development of the method and drafting the manuscript. R.Á. implemented the method and the visualisation of the results. A.M. consulted on data availability and reviewed the manuscript. P.H. and M.L. conceived the research question and provided input on the clinical parameters and interpretation. The CoMo Consortium contributed to discussions on the underlying conceptual approach and methodology used and reviewed the manuscript.

## Competing interests

The authors declare no competing interests.

## Additional information

## the CoMo Consortium

Fatima Arifi[6], Chynar Zhumalieva[7], Inke N. D. Lubis[8], Antoninho Benjamin Monteiro[9], Ainura Moldokmatova[3], Siyu Chen[3], Aida Estebesova[10], Mofakhar Hussain[11], Dipti Lata[12], Emmanuel A. Bakare[13], Biniam Getachew[14], Mohammad Nadir Sahak[6], Phetsavanh Chanthavilay[15], Akindeh M. Nji[16], Yu Nandar Aung[17], Nathaniel Hupert[18], Sai Thein Than Tun[1], Wirichada Pan-Ngum[19], K. C. Sarin[20], Handoyo Harsono[21], Sana Eybpoosh[22], Renato Mendes Coutinho[23], Semeeh A. Omoleke[24], Amen-Patrick Nwosu[25], Nantasit Luangasanatip[19], Ainura Kutmanova[26], Aizhan Dooronbekova[7], Antonio Ximenes[27], Merita Monteiro[28], Olivier Celhay[29], Keyrellous Adib[30], Amel H. Salim[31], Yuki Yunanda[32], Mahnaz Hossain Fariba[33], Amirah Azzeri[34,35], Penny Hancock[3], Hakim Bekrizadeh[36], Sayed Ataullah Saeedzai[37], Ivana Alona[32], Grace Wezi Mzumara[38], Joao Martins[27], Jose L. Herrera-Diestra[39], Hamid Sharifi[40], Talant Abdyldaev[41], Babak Jamshidi[42], Noran Naqiah Hairi[43], Naima Nasir[4], Rashid U. Zaman[44], Sopuruchukwu Obiesie[45], Roberto A. Kraenkel[46], Nicholas Letchford[47], Lucsendar Raimunda Fernandes Alves[28], Sandra Adele[25], Lorena Suárez-Idueta[48], Nicole Advani[25], Manar Marzouk[49], Viviana Mabombo[25], Aibek Mukambetov[50], Adeniyi Kolade Aderoba[4], Bpriya Lakshmy Tbalasubramaniam[25], Nicole Feune de Colombi[4], Maria Angela Varela Niha[51], Francisco Obando[52], Parinda Wattanasri[4], Sompob Saralamba[19], Fatiha Shabaruddin[53], Shafiun Nahin Shimul[54], Maznah Dahlui[55], Reshania Naidoo[4], Caroline Franco[46], Michael G. Klein[56], Aisuluu Kubatova[7], Nusrat Jabin[25], Shwe Sin Kyaw[1], Luzia Tomas Freitas[4], Sunil Pokharel[25], Proochista Ariana[4], Chris Erwin G. Mercado[57], Shamil Ibragimov[50], John Robert C. Medina[58] & Mesulame Namedre[59]

[6]World Health Organization, Kabul, Afghanistan. [7]Public Fund "Institution of social development" in the Kyrgyz Republic, Bishkek, Kyrgyz Republic. [8]Department of Paediatrics, Faculty of Medicine, Universitas Sumatera Utara, Medan, Indonesia. [9]Universidade da Paz, Dili, Timor-Leste. [10]USAID Mission in the Kyrgyz Republic, Bishkek, Kyrgyz Republic. [11]Institute of Health Policy, Management and Evaluation, University of Toronto, Toronto, Canada. [12]Oxford Policy Management, Lalitpur, Nepal. [13]Department of Mathematics, Federal University, Oye-Ekiti, Ekiti State, Nigeria. [14]Stop Transmission of Polio Program (STOP) program, World Health Organization, Birnin Kebbi, Nigeria. [15]Institute of Research and Education Development, University of Health Sciences, Vientiane, Lao PDR. [16]Laboratory for Public Health Research biotechnology, Biotechnology Center, University of Yaounde, Yaounde, Cameroon. [17]London School of Economics, London, UK. [18]Department of Medicine, Weill Cornell Medicine, and Cornell Institute for Disease and Disaster Preparedness, Cornell University, Ithaca, USA. [19]Mathematical And Economic MODelling Group, Mahidol-Oxford Tropical Medicine Research Unit, Faculty of Tropical Medicine, Mahidol University, Bangkok, Thailand. [20]Health Intervention and Technology Assessment Program (HITAP), Ministry of Public Health, Nonthaburi, Thailand. [21]COVID-19 Task Force for North Sumatera province,

Medan, Indonesia. [22]Department of Epidemiology and Biostatistics, Research Centre for Emerging and Reemerging Infectious Diseases, Pasteur Institute of Iran, Tehran, Iran. [23]Center for Mathematics, Computation and Cognition, Federal University of ABC, São Paulo, Brazil. [24]World Health Organization, Birnin Kebbi, Nigeria. [25]Nuffield Department of Medicine, University of Oxford, Oxford, UK. [26]Ministry of Public Health of the Kyrgyz Republic, Bishkek, Kyrgyz Republic. [27]Universidade Nacional Timor Lorosae, Dili, Timor-Leste. [28]Menzies School of Health Research, Charles Darwin University, Darwin, Australia. [29]Independent researcher (no affiliation), Ithaca, NY, USA. [30]University of Nottingham, Nottingham, UK. [31]Population Services International, Addis Ababa, Ethiopia. [32]Department of Community Medicine, Faculty of Medicine, Universitas Sumatera Utara, Medan, Indonesia. [33]Government of People's Republic of Bangladesh, Dhaka, Bangladesh. [34]Department of Research, Development and Innovation, University of Malaya Medical Centre, Kuala Lumpur, Malaysia. [35]Department of Primary Care, Faculty of Medicine and Health Sciences, Universiti Sains Islam Malaysia (USIM), Kuala Lumpur, Malaysia. [36]Department of Statistics, Payame Noor University, Tehran, Iran. [37]Ministry of Public Health, Kabul, Afghanistan. [38]Malawi Ministry of Health, Lilongwe, Malawi. [39]Centro de Simulacion y Modelos (CeSiMo), Universidad de Los Andes, Merida, Venezuela. [40]HIV/STI Surveillance Research Center, and WHO Collaborating Center for HIV Surveillance, Institute for Futures Studies in Health, Kerman University of Medical Sciences, Kerman, Iran. [41]Medical Association in the Kyrgyz Republic, Bishkek, Kyrgyz Republic. [42]Department of Biostatistics, Kermanshah University of Medical Sciences, Kermanshah, Iran. [43]University of Malaya, Kuala Lumpur, Malaysia. [44]Oxford Policy Management, Oxford, UK. [45]Centre for Tropical Medicine and Global Health, Nuffield Department of Medicine, University of Oxford, Oxford, UK. [46]Institute for Theoretical Physics, São Paulo State University (UNESP), São Paulo, Brazil. [47]Oxford Policy Management, Oxford, UK. [48]University College of London, London, UK. [49]Valid International Ltd, Oxford, UK. [50]Soros Foundation in the Kyrgyz Republic, Bishkek, Kyrgyz Republic. [51]Ministry of Health Timor-Leste, Dili, Timor-Leste. [52]PEAK Urban and Oxford Martin Informal Cities Programmes, COMPAS, School of Anthropology and Museum Ethnography, University of Oxford, Oxford, UK. [53]Faculty of Pharmacy, University of Malaysia, Kuala Lumpur, Malaysia. [54]Institute of Health Economics, University of Dhaka, Dhaka, Bangladesh. [55]Department of Social and Preventive Medicine, Faculty of Medicine, University of Malaya, Kuala Lumpur, Malaysia. [56]Department of Marketing & Business Analytics, Lucas College and Graduate School of Business, San José State University, San José, USA. [57]Independent researcher (no affiliation), Manila, Philippines. [58]Department of Epidemiology and Biostatistics, College of Public Health, University of the Philippines Manila, Manila, Philippines. [59]Fiji Center for Disease Control (CDC), Suva, Fiji.

