## [Peer Review File · Nature Communications]

REVIEWER COMMENTS

Reviewer #1 (Remarks to the Author):

This is an interesting paper. some remarks apply:

I think for the extend of the analysis the evidence on the effectiveness of dexamethasone is weak, I think one ref only, cannot this be strengthened?

Effectiveness will differ between countries, in particular between HICs and LMICs...

The paper seems to be on UK and then global... the use of Spanish and French data is bit confusing

There seems to be a disbalance between the straightfwd calculus in the methods and the complex math formulations in the Suppl materials.

Reviewer #2 (Remarks to the Author):

This is a well written carefully argued paper that suggests that dexamethasone is value for money in the UK in the treatment of patients with clinically suspected or laboratory confirmed SARS-CoV-2 infection who receive respiratory support.

The paper goes on to argue that those without access to supplementary oxygen can also benefit at an acceptable cost. The first claim is not contentious as the NEJM trial demonstrates a significant reduction in mortality in group with an average life expectancy of around 19 years, a per patient cost of a course of dexamethasone that is less than \$10 in many countries, and some potential additional costs from increased hospital length of stay outside of ICU for survivors.

Modelling beyond the trial and predicting effectiveness in practice across different jurisdictions inevitably involves assumptions about effectiveness and the baseline mortality risk and the authors seem to make plausible assumptions at least for some countries. As they note the trial evidence does not support an effect of dexamethasone among patients who were not receiving any respiratory support at randomization and so any extrapolation is based on alternative best available evidence.

The approach is to make a plausible assumption about the mortality rate and the protective effectiveness of dexamethasone in this group but to look at two bounds as sensitivity analyses (the same as if the person received optimal oxygen or no effect)

In estimating the cost effectiveness of dexamethasone in the UK the authors are hampered by the lack of accurate data on proportion of patients by age who are hospitalised and the case fatality rate in the UK. The therefore rely on French and Spanish estimates.

It might be helpful to avoid this kind of assumption initially if the authors estimate of the cost effectiveness of dexamethasone based solely on the UK trial prior to further modelling. It might look something like:

As the authors have calculated from the trial: an 18% reduction in death in mech vent from dexamethasone and 36% in non ventilated if of 100,000 hospitalised 40000 died without then it would save $[(0.18 \times 0.59) + (0.36 \times 0.17)] \times 40,000 = 6,696$. They would have some more days in hospital say 5 more at a cost of $6,696 \times 1350$ pounds and perhaps ignoring a few less in ICU (and any reduction in los for those who would not have died) with drug cost of 5 pounds for each of those treated (160,000) then the total cost is under 10m pounds so cost per life year saved of $10m/24.2 \times 6696$ or about \$60 per life year saved

In modelling beyond the circumstances where oxygen therapy is available the authors are limited by the trial evidence so far where there was no clear effect of dexamethasone among patients who were not receiving any respiratory support at randomization (17.8% vs. 14.0%; rate ratio, 1.19; 95% CI, 0.91 to 1.55) "

The reasonable approach in the paper is to either assume no effect or to look at two bounds (no effect and the same as if the person received optimal oxygen). (1) dexamethasone has the same relative impact on mortality in patients who do not receive the respiratory support they require as in those who do; (2) dexamethasone has no impact on patient outcome if the patient does not receive the oxygen or ventilation they need. There is quite a large gain assumed under the first bound presumably because as the authors say "it is reasonable to assume that some mortality benefit would arise from the reduction of inflammation and consequent decrease in the probability of progressing to respiratory failure." This seems plausible and while I don't have the clinical knowledge to assess its validity I would like to see stronger justification.

There is some uncertainty around the effect of dexamethasone on mortality for a) patients who need oxygen but do not receive it and b) patient on oxygen who transit to mechanical ventilation. The paper assumes that 74% of patients need oxygen and the probability of an hospitalised patient who receives oxygen needing a ventilator if on dexamethasone is between 0.22 and 0.87. Reduction in mortality for patients requiring non-invasive oxygen if on dexamethasone is 0.2 in table 1 (0.18 in the trial) and the reduction in mortality for patients requiring a ventilator if on dexamethasone is 0.36 (as in the trial). It is not clear in the text however how the authors classify patients and attribute risk reductions to dexamethasone for those who receive non invasive oxygen who would have progressed to mechanical ventilation and subsequently died but no longer do so (pvd). I am not clear if there is evidence on this group, and while the base assumption seems to be no difference, somewhere there is an assumption of a lower transition probability for the dexamethasone group, but what appears to be the same mortality reduction for all those who are on mechanical ventilation irrespective of prior history of treatment. The authors could make this clearer.

Reviewer #3 (Remarks to the Author):

Thank you for giving us the opportunity to review this important manuscript.

We have two sets of comments on this piece, one, on the methods to assess the economic impact of the intervention and the other, on the policy implications of this analysis. These are elaborated below:

Economic methods

- The study appears to underestimate costs incurred for the introducing dexamethasone in the treatment of COVID-19 patients in hospitals by omitting the cost of providing COVID-19 treatment if patients live longer and require the full course of treatment. As per standard methods, the direct medical cost of additional days for the group that survives on account of dexamethasone should be included. Without these costs, the overall cost for the intervention will likely be underestimated.
- On the other hand, the study overestimates the benefits because it does not discount the life years gained. The cost of treatment should not be discounted given the short period considered, but since the outcomes of life years gained is for a longer period, these should be discounted. This is particularly important since the authors compare the Incremental Cost-effectiveness Ratio (ICER) to the threshold of NICE in the UK which specifies, in its guidelines, that future benefits should be discounted.
- The underestimation of costs and overestimation of benefits makes the ICER appear to be

extremely good i.e. highly cost-effective. That said, we don't expect that the conclusion of the study –the use of dexamethasone to treat COVID-19 patients is a very cost-effective intervention - will change once the points on costs and outcomes are accounted for.

Policy applications

- The study shows that dexamethasone can be cost-effective in treating COVID-19 patients if it is provided in the UK where there is no problem of oxygen access. However, the results are not known for countries where the access to oxygen is limited, a research area for future studies, as pointed out by the authors. This has policy implications for the transferability of the study results, which are also mentioned by the author and regarding which we would like to suggest the following:

- This study should be able to inform design of clinical studies in low-and-middle income countries (LMICs). This could be done using value of information analysis to inform which model parameters are of importance if this policy is to be used in another setting with a much lower cost-effectiveness threshold, called Expected Value of Partial Perfect Information (EVPPPI), or, even to determine how many samples for each arm of the future clinical study in LMICs will be needed, with different assumptions of the accessibility to oxygen and difference in mortality of the two groups, called the expected value of sample information (EVSI). This can be an important contribution to the understanding of the intervention as the effectiveness of introducing dexamethasone in treating COVID-19 patients may be different in different settings. For an example to highlight the importance of settings in determining clinical effectiveness, consider the effect of oral rehydration, a simple intervention which may work wonders for treating diarrhea but one that is not a feasible option for treating patients in settings where access to clean water is lacking.

The authors would like to thank the reviewers for their insightful and helpful comments. We respond to each point below.

Reviewer #1 (Remarks to the Author):

Comment:

this is an interesting paper. some remarks apply:

I think for the extend of the analysis the evidence on the effectiveness of dexamethasone is weak, I think one ref only, cannot this be strengthened?

Response:

We chose to use the RECOVERY study as this was the only study which provided an effect size of dexamethasone stratified by non-invasive and invasive oxygen. Indeed other trials have been published (two of which are described in this meta-analysis*), but those studies they only recruited patients requiring invasive oxygen. As presented in the primary RECOVERY paper and highlighted in the manuscript under consideration here, the effect size of dexamethasone on patients requiring either non-invasive or invasive oxygen are significantly different, and thus were taken into account in our analysis. However, the effect size for this patient group in the RECOVERY trial does lie within the confidence intervals of the other trials. The reference to the meta-analysis paper has been included in the main text (*line 32*).

* The WHO Rapid Evidence Appraisal for COVID-19 Therapies (REACT) Working Group. Association Between Administration of Systemic Corticosteroids and Mortality Among Critically Ill Patients With COVID-19: A Meta-analysis. JAMA. Published online September 02, 2020. doi:10.1001/jama.2020.17023

Comment:

Effectiveness will differ between countries, in particular between HICs and LMICs...

Response:

We agree with the reviewer on this and believe that the main divergence will be due to the access to oxygen treatment. The potential effect of dexamethasone in patients who cannot access non-invasive or invasive oxygen when they require it is a source of huge uncertainty. We discuss this issue on *lines 106-126*. We estimated the potential number of lives saved under two assumptions: (1) dexamethasone has the same relative impact on mortality in patients who do not receive the respiratory support they require as in those who do; (2) dexamethasone has no impact on patient outcome if the patient does not receive the oxygen or ventilation they need. We also highlight this issue in the abstract with specific reference to LMICs.

Comment:

The paper seems to be on UK and then global... the use of Spanish and French data is bit confusing

Response:

We use an infection severity (i.e. requiring hospital) rate (ihr) by age combined with severity fatality rate (hfr) by age which is modified by the level of access to hospital, non-invasive O2 and ventilators. The ihr has rarely been measured since it requires the appropriate data for the denominator, which is all infections (including asymptomatic ones). This information is not readily available, so we have therefore used the few vectors which have been published for the European context, namely for Spain and France.

The following text on *lines 332-334* articulates this decision: “Ideally, we would use UK age-dependent serology data to inform a UK specific IHR. Unfortunately, those data are not currently available so we must rely on other countries estimates/data.”

Comment:

There seems to be a disbalance between the straightfwd calculus in the methods and the complex math formulations in the Suppl materials.

Response:

We assume that the reviewer is comparing the retrospective estimate of lives which could have been saved in the UK shown on *line 65* with the prospective analysis in the main methods section of the manuscript. We agree that the mathematics are a little more involved for the prospective analysis. We believe this is justified since we must consider uncertainty in projections forward in time since the exposure may be at a variety of levels compared with the retrospective analysis with a fixed and measured exposure. This is reflected in the pessimistic range of explored future attack rates, accounting for a potentially worse second wave during the UK winter, in the absence of vaccination. Given this and one other comment by Reviewer 2, we decided to refrain from explicitly calculate numbers of deaths saved with the simple formula, and rather present a weighted mortality reducing risk for dexamethasone in a hospital setting. We thus hope to ensure that the analysis remains transparent and reproducible.

Reviewer #2 (Remarks to the Author):

Comment:

This is a well written carefully argued paper that suggests that dexamethasone is value for money in the UK in the treatment of patients with clinically suspected or laboratory confirmed SARS-CoV-2 infection who receive respiratory support.

The paper goes on to argue that those without access to supplementary oxygen can also benefit at an acceptable cost. The first claim is not contentious as the NEJM trial demonstrates a significant reduction in mortality in group with an average life expectancy of around 19 years, a per patient cost of a course of dexamethasone that is less than \$10 in many countries, and some potential additional costs from increased hospital length of stay outside of ICU for survivors.

Modelling beyond the trial and predicting effectiveness in practice across different jurisdictions inevitably involves assumptions about effectiveness and the baseline mortality risk and the authors seem to make plausible assumptions at least for some countries. As they note the trial evidence does not support an effect of dexamethasone among patients who were not receiving any respiratory support at randomization and so any extrapolation is based on alternative best available evidence. The approach is to make a plausible assumption about the mortality rate and the protective effectiveness of dexamethasone in this group but to look at two bounds as sensitivity analyses (the same as if the person received optimal oxygen or no effect)

Response:

The authors thank the reviewer for this accurate and positive summary of our approach.

Comment:

In estimating the cost effectiveness of dexamethasone in the UK the authors are hampered by the lack of accurate data on proportion of patients by age who are hospitalised and the case fatality rate in the UK. The therefore rely on French and Spanish estimates.

Response:

We use an infection severity (i.e. requiring hospital) rate (ihr) by age combined with severity fatality rate (hfr) by age which is modified by the level of access to hospital, non-invasive O2 and ventilators. The ihr has rarely been measured since it requires the appropriate data for the denominator, which is all infections (including asymptomatic ones). This information is not readily available, so we have therefore used the few vectors which have been published for the European context, namely for Spain and France. The following text on *lines 332-334* articulates this decision: “Ideally, we would use UK age-dependent serology data to inform a UK specific IHR. Unfortunately, those data are not currently available so we must rely on other countries estimates/data.”

Comment:

It might be helpful to avoid this kind of assumption initially if the authors estimate of the cost effectiveness of dexamethasone based solely on the UK trial prior to further modelling. It might look something like:

As the authors have calculated from the trial: an 18% reduction in death in mech vent from dexamethasone and 36% in non-ventilated if of 100,000 hospitalised 40,000 died without then it would save $[(0.18 \times 0.59) + (0.36 \times 0.17)] \times 40,000 = 6,696$. They would have some more days in hospital say 5 more at a cost of $6,696 \times 1350$ pounds and perhaps ignoring a few less in ICU (and any reduction in los for those who would not have died) with drug cost of 5 pounds for each of those treated (160,000) then the total cost is under 10m pounds so cost per life year saved of $10m/24.2 \times 6696$ or about \$60 per life year saved

Response:

We appreciate the reviewer’s comments and admit we did struggle with the notion of potentially offering a full retrospective analysis, providing a cost-effectiveness metrics applied to the mortality rates observed so far. We prefer to use an alternative approach, where we explore a range of probable attack rates given that the winter wave is expected to be slightly higher than the April one, in the absence of a vaccine. In *Figure 2* it is clear that the 40,000 deaths represent the lower end of the IQR. Following that, we observe that the \$60 per life year saved number suggested is in line with the lower end for the ICER in *Figure 2*. However, we feel that providing any metric of cost-effectiveness to reflect the potential miss-use of non-deployment of a health intervention during the initial pandemic period is miss-guided since the study conclusively showing the potential of the drug was only published in July, and also too politically charged. We would rather focus on exploring the future benefits of treating COVID-19 patients with Dexamethasone and whether that is cost effective. To avoid getting into the issue of numbers of lives saved and potential costs of those, we have changed this framing of the issue paragraph slightly. It now reflects only that dexamethasone can reduce deaths by 16.75% if 59% of patients receive oxygen and 17% are ventilated.

Comment:

The reasonable approach in the paper is to either assume no effect or to look at two bounds (no effect and the same as if the person received optimal oxygen). (1) dexamethasone has the same relative impact on mortality in patients who do not receive the respiratory support they require as in those who do; (2) dexamethasone has no impact on patient outcome if the patient does not receive the oxygen or ventilation they need. There is quite a large gain assumed under the first bound presumably because as the authors say “it is reasonable to assume that some mortality benefit would arise from the reduction of inflammation and consequent decrease in the probability of progressing to respiratory failure.” This seems plausible and while I don’t have the clinical knowledge to assess its validity I would like to see

stronger justification.

Response:

We agree that we have made a strong enough case for the plausibility of the upper bound. As we mention in the manuscript in *lines 114-118*, there is no data to inform the potential impact of dexamethasone on patients who require but cannot access supplemental oxygen or invasive mechanical ventilation. As such we employed a completely agnostic approach and explored the limits of plausibility, i.e., assuming that dexamethasone either has the same impact or no impact at all. The true impact of dexamethasone on such patients is likely to fall within the interval provided here. We expanded the text to better frame this approach.

Comment:

There is some uncertainty around the effect of dexamethasone on mortality for a) patients who need oxygen but do not receive it and b) patient on oxygen who transit to mechanical ventilation. The paper assumes that 74% of patients need oxygen and the probability of an hospitalised patient who receives oxygen needing a ventilator if on dexamethasone is between 0.22 and 0.87. Reduction in mortality for patients requiring non-invasive oxygen if on dexamethasone is 0.2 in table 1 (0.18 in the trial) and the reduction in mortality for patients requiring a ventilator if on dexamethasone is 0.36 (as in the trial). It is not clear in the text however how the authors classify patients and attribute risk reductions to dexamethasone for those who receive non invasive oxygen who would have progressed to mechanical ventilation and subsequently died but no longer do so (pvd). I am not clear if there is evidence on this group, and while the base assumption seems to be no difference, somewhere there is an assumption of a lower transition probability for the dexamethasone group, but what appears to be the same mortality reduction for all those who are on mechanical ventilation irrespective of prior history of treatment. The authors could make this clearer.

Response:

We have two categories of patient who are eligible for dexamethasone treatment. Those who require non-invasive oxygen (but never require ventilation) and those who ultimately require mechanical ventilation (where we assume their use of non-invasive oxygen during this pathway is negligible in terms of time and cost). Each group has two survival rates associated with them depending on whether they receive dexamethasone treatment or not, as reported as the primary outcome in the RECOVERY paper. The secondary outcome of the RECOVERY paper indicates that 13% of patients will not progress to requiring ventilation if they receive dexamethasone treatment. This figure is used to modify the number of patients who would be in the ventilator group. Text has been added to the legend of figure 1 to clarify this.

Reviewer #3 (Remarks to the Author):

Thank you for giving us the opportunity to review this important manuscript.

We have two sets of comments on this piece, one, on the methods to assess the economic impact of the intervention and the other, on the policy implications of this analysis. These are elaborated below:

Economic methods

Comment:

The study appears to underestimate costs incurred for the introducing dexamethasone in the treatment of COVID-19 patients in hospitals by omitting the cost of providing COVID-19 treatment if patients live longer and require the full course of treatment. As per standard methods, the direct medical cost of additional days for the group that survives on account of dexamethasone should be included. Without these costs, the overall cost for the intervention will likely be underestimated.

Response:

Thank you for this comment. We have already included an extra 4.5 days as the incremental number of days spent in hospital by survivors (supplementary table 2). With respect to an increased cost of dexamethasone treatment, a discussion with the RECOVERY trial authors ascertained that dexamethasone was administered to patients, for a median of 7 days [IQR 3-10 days]. However, no records were kept on the duration of dexamethasone treatment in those who survived vs those that died. Nevertheless, the cost of dexamethasone itself (~\$10 per full treatment course) is negligible compared to the cost of a night spent at the hospital (over \$1,500 for a single night), so the direct medical costs of survivorship are dominated by the later (which is already included in the analysis as mentioned above).

Comment:

On the other hand, the study overestimates the benefits because it does not discount the life years gained. The cost of treatment should not be discounted given the short period considered, but since the outcomes of life years gained is for a longer period, these should be discounted. This is particularly important since the authors compare the Incremental Cost-effectiveness Ratio (ICER) to the threshold of NICE in the UK which specifies, in its guidelines, that future benefits should be discounted.

Response:

Thank you for this comment. There has been a lot of debate amongst economists about the value of discounting benefits as measured by life-years gained. Nevertheless, we have taken the reviewers' comments into consideration and have applied a discount rate of 1.5% to weigh each life year gained as recommended by NICE and updated the figures, tables and text accordingly.

Comment:

The underestimation of costs and overestimation of benefits makes the ICER appear to be extremely good i.e. highly cost-effective. That said, we don't expect that the conclusion of the study –the use of dexamethasone to treat COVID-19 patients is a very cost-effective intervention - will change once the points on costs and outcomes are accounted for.

Response:

This is addressed in the two points above. As the reviewer rightly observed, while the ICER changed somewhat, the conclusion of the study that the use of dexamethasone to treat COVID-19 patients is a cost-effective intervention remained unchanged.

Policy applications

- The study shows that dexamethasone can be cost-effective in treating COVID-19 patients if it is provided in the UK where there is no problem of oxygen access. However, the results are not known for countries where the access to oxygen is limited, a research area for future studies, as pointed out by the authors. This has policy implications for the transferability of the study results, which are also mentioned by the author and regarding which we would like to suggest the following:
 - This study should be able to inform design of clinical studies in low-and-middle income countries (LMICs). This could be done using value of information analysis to inform which model parameters are of importance if this policy is to be used in another setting with a much lower cost-effectiveness threshold, called Expected Value of Partial Perfect Information (EVPPi), or, even to determine how many samples for each arm of the future clinical study in LMICs will be needed, with different assumptions of the accessibility to oxygen and difference in mortality of the two groups, called the expected value of sample information (EVSI). This can be an important contribution to the understanding of the intervention as the effectiveness of introducing dexamethasone in treating COVID-19 patients may be

different in different settings. For an example to highlight the importance of settings in determining clinical effectiveness, consider the effect of oral rehydration, a simple intervention which may work wonders for treating diarrhea but one that is not a feasible option for treating patients in settings where access to clean water is lacking.

Response:

We agree with the comments of the reviewer and have added some language as next steps/recommendations in the discussion section (*lines 159:162*).

REVIEWER COMMENTS

Reviewer #2 (Remarks to the Author):

I am satisfied that the authors have addressed the comments that I made within the limits of the data

Reviewer #3 (Remarks to the Author):

We thank the authors for making changes to the manuscript in light of our comments. Originally, we had requested the authors to conduct Expected Value of Partially Perfect Information (EVPPI) to identify priority parameters for future clinical studies, with the hope that we will also be able to see cost-effectiveness acceptability curves (CEAC) of the analysis.

CEAC provides information on the probability of the interventions under investigation being good value for money at different levels cost-effectiveness thresholds, taking into account parameter uncertainty. This is based on probabilistic modelling, which is a standard methodological approach for economic evaluations and is included in the CHEERS guideline (link: <https://www.equator-network.org/reporting-guidelines/cheers/>)

While we accept that the authors discuss the potential use of EVPPI to inform future clinical studies, we urge them to at least construct a CEAC. This analysis will provide useful information to readers in different countries, each of which will likely have different cost-effectiveness thresholds, to make health resource allocation decisions.

The authors would like to thank the reviewers for their review and insightful comments. We respond to each point below.

Reviewer #2 (Remarks to the Author):

Comment:

Reviewer #2 (Remarks to the Author):

I am satisfied that the authors have addressed the comments that I made within the limits of the data

Response:

The authors thank the reviewer for the concurrence with the revised submission.

Reviewer #3 (Remarks to the Author):

We thank the authors for making changes to the manuscript in light of our comments. Originally, we had requested the authors to conduct Expected Value of Partially Perfect Information (EVPPPI) to identify priority parameters for future clinical studies, with the hope that we will also be able to see cost-effectiveness acceptability curves (CEAC) of the analysis. CEAC provides information on the probability of the interventions under investigation being good value for money at different levels cost-effectiveness thresholds, taking into account parameter uncertainty. This is based on probabilistic modelling, which is a standard methodological approach for economic evaluations and is included in the CHEERS guideline (link: <https://www.equator-network.org/reporting-guidelines/cheers/>). While we accept that the authors discuss the potential use of EVPPPI to inform future clinical studies, we urge them to at least construct a CEAC. This analysis will provide useful information to readers in different countries, each of which will likely have different cost-effectiveness thresholds, to make health resource allocation decisions.

Response:

The authors thank the reviewer for the thoughtful review and the recommendation to construct a cost-effectiveness acceptability curve (CEAC). We have made the appropriate changes to the manuscript to reflect the reviewer's recommendation. A cost-effectiveness acceptability curve (CEAC) was constructed to quantify and graphically represent the uncertainty surrounding the adoption of dexamethasone at various a ICEAR thresholds. The findings show that at a maximum acceptable ceiling ratio of £2,020 per life year gained, the decision uncertainty is 5% (given the probability that dexamethasone is cost-effective is 0.95) (Figure 3A).

We have added the description of the methods in lines 357-378. In addition, the findings are discussed in the main text (lines 145-149) including text discussing the uncertainty inherent to assumptions on access to oxygen therapy and the efficacy of dexamethasone in patients that do not receive oxygen. We add a recommendation, based on the results presented in Figure 3B, that clinical studies in settings expected to have much lower access to oxygen therapy should be considered as a next step to understand the effectiveness of introducing dexamethasone in different settings (lines 171-176). We state that those studies should be informed by a comprehensive expected value of partially perfect information (EVPPPI) analysis extending on the results presented here to inform the most appropriate sample sizes, as we consider this analysis to be outside of the remit of the current submission.

We look forward to a favorable response.

REVIEWERS' COMMENTS

Reviewer #3 (Remarks to the Author):

Thank you for addressing the points we raised. We have no further comments.